# Metabolome and Transcriptome Profiling Reveal Carbon Metabolic Flux Changes in *Yarrowia lipolytica* Cells to Rapamycin

**DOI:** 10.3390/jof8090939

**Published:** 2022-09-06

**Authors:** Ziyu Liu, Junjie Tian, Zhengang Miao, Wenxing Liang, Guangyuan Wang

**Affiliations:** 1Shandong Province Key Laboratory of Applied Mycology, College of Life Sciences, Qingdao Agricultural University, Qingdao 266109, China; 2Shandong Engineering Research Center for Environment-Friendly Agricultural Pest Management, College of Plant Health and Medicine, Qingdao Agricultural University, Qingdao 266109, China

**Keywords:** *Yarrowia lipolytica*, rapamycin, TOR, lipids, amino acid, nitrogen-rich, metabolomics, transcriptomics

## Abstract

*Yarrowia lipolytica* is an oleaginous yeast for the production of oleochemicals and biofuels. Nitrogen deficiency is beneficial to lipids biosynthesis in *Y. lipolytica*. Target of rapamycin (TOR) regulates the utilization of nutrients, which is inhibited in nitrogen starvation or by rapamycin treatment. However, under nitrogen-rich conditions, the lipids biosynthesis in *Y. lipolytica* after inhibition of TOR by rapamycin is elusive. Combining metabolomics and transcriptomics analysis, we found that rapamycin altered multiple metabolic processes of *Y. lipolytica* grown in nitrogen-rich medium, especially the metabolisms of amino acids and lipids. A total of 176 differentially accumulated metabolites were identified after rapamycin treatment. Rapamycin increased the levels of tryptophan, isoleucine, proline, serine, glutamine, histidine, lysine, arginine and glutamic acid, and decreased the levels of threonine, tyrosine and aspartic acid. Two fatty acids in lipid droplets, stearic acid (down-regulated) and stearidonic acid (up-regulated), were identified. The expression of 2224 genes changed significantly after rapamycin treatment. Further analysis revealed that rapamycin reduced carbon flux through lipids biosynthesis, accompanied by increased carbon flux through fatty acids degradation and amino acid (especially glutamic acid, glutamine, proline and arginine) biosynthesis. The dataset provided here is valuable for understanding the molecular mechanisms of amino acid and lipids metabolisms in oleaginous yeast.

## 1. Introduction

In recent years, utilization of oleaginous yeasts as sources of oils has aroused great interest in the fermentation industry. Microbial oils produced by oleaginous yeasts have a wide range of applications in the field of biotechnology, e.g., as a feedstock for biodiesel production [1], for the production of fatty alcohols, alkanes, conjugated linoleic acid, and long chain polyunsaturated fatty acids [2], and replacing high value-added cocoa butter [3]. Oleaginous microorganisms are those that can accumulate intracellular lipids up to 20% of their cell dry weight [4]. *Yarrowia lipolytica* is an aerobic oleaginous yeast [5]. It has been well documented that *Y. lipolytica* is a good candidate for production of oleochemicals and drop-in transportation fuels because of its ability to biosynthesize significant quantities of lipids in cells [6].

Biosynthesis of microbial oils usually requires nitrogen-starvation conditions [7]. Nitrogen limitation first induces citric acid (CA) accumulation in the cells of oleaginous microorganisms. Subsequently, the accumulated CA is split into acetyl-CoA (AcCoA) and oxaloacetic acid (OAA) by ATP-citrate lyase (ACL) [1]. The resulting AcCoA is used as a critical feedstock for lipids biosynthesis by oleaginous yeasts. AcCoA is converted into long-chain fatty acids by the catalysis of AcCoA carboxylase (ACC) and fatty acid synthase complex (FAS1 and FAS2) [8].

Since nitrogen restriction favors lipids accumulation in oleaginous microorganisms, it is easy to speculate that the target of rapamycin (TOR) pathway, an established pathway for nitrogen sensing and discrimination [9,10,11], is involved in the regulation of lipids metabolism. TOR is activated under high-nutrient conditions, but inhibited under nitrogen starvation or by rapamycin treatment [11]. Since nitrogen limitation inhibits TOR activity [11], this suggests that reducing TOR activity is beneficial for lipids biosynthesis in oleaginous microorganisms. However, it is not clear whether the lipids level of oleaginous yeasts can be increased by inhibiting TOR under nitrogen-rich conditions.

Interestingly, in the condition of a nitrogen-rich source, inhibition of TOR by rapamycin could result in a fast lipid-droplet replenishment in *Saccharomyces cerevisiae* [12]. A lipid droplet is an intracellular structure, which is the core of neutral lipids, and the main components in lipid droplets are triacylglycerols (TAGs) and sterol esters (SEs) [13,14]. Oleaginous yeasts can biosynthesize more intracellular oil droplets than *S. cerevisiae*. Our previous studies have shown that many oily yeasts, such as *Y. lipolytica* [15], *Rhodotorula taiwanensis* [1], and *Aureobasidium pullulans* var. *aubasidani* [16], biosynthesized oil droplets that could fill the entire cell. Similar to those of *S. cerevisiae*, TAGs and SEs are also the main components in the lipid droplets accumulated in oleaginous yeasts [2,7].

While in *S. cerevisiae* there are two different TOR kinases [12], the oleaginous yeast *Y. lipolytica* owns only a single TOR kinase [17]. Although inhibition of TOR under nitrogen-rich conditions could promote lipids biosynthesis in *S. cerevisiae* [12], it is unclear whether this also applies to oleaginous yeast.

In this study, to investigate TOR’s regulation in metabolism of the oleaginous yeast *Y. lipolytica* under a rich nitrogen environment, we performed a multiple omics analysis using metabolomics and transcriptomics. In total, we identified 176 differentially accumulated metabolites and 2224 differentially expressed genes. These metabolites and genes are involved in various intracellular metabolic pathways such as amino acid biosynthesis and lipids metabolism. More importantly, we found that rapamycin lowered carbon flux through lipids biosynthesis while increasing carbon flux through fatty acid breakdown and amino acid production (particularly glutamic acid, glutamine, proline, and arginine). These findings provide a new insight in the understanding of how carbon flux transition is regulated in response to rapamycin by the oleaginous yeast *Y. lipolytica*.

## 2. Materials and Methods

### 2.1. Strains and Growth Assays

The yeast strain 2E00681 of *Y. lipolytica* used in this study [18,19], an uracil mutant isolated from the yeast strain of ACA-DC 50109 (wild-type) [20] using 5′-fluororotic acid [21], was kindly offered by Prof. Zhen-Ming Chi at Ocean University of China. *Y. lipolytica* 2E00681 was kept 4 °C on YPD agar slant (1% yeast extract, 2% peptone, 2% dextrose, and 1.5% agar). For growth assays, after being grown in YPD liquid medium (50 mL in 250-mL shake-flask) at 28 °C for 12 h, the cells in the culture were diluted to OD_600_ = 0.25 using fresh YPD liquid medium. Then, 2 μL of yeast seed was plated on the YPD agar plates containing rapamycin from 50 to 100 ng/mL, followed by inoculation at 28 °C for 24 h. The resulting colonies along with agar were transferred to 2.0 mL of 10 mM PBS (pH 7.0), respectively. After vortex for 2 min, the number of cells in the suspensions was determined using a hemocytometer (XB-K-25) under a microscope.

### 2.2. Intracellular Lipids Fluorescence Intensity

The yeast cells grown on the YPD agar plates containing rapamycin were harvested as described in the above section and washed using 10 mM PBS (pH 7.0) by centrifugation at 9000× *g* for 5 min, followed by resuspension in PBS with cell density OD_600_ = 1.0. A total of 200 μL of cells suspension was supplemented with 4.0 μL of 0.1 mg/mL Nile red in DMSO. After being stained for 5 min in the dark at room temperature, fluorescence was read at 530/590 nm using a 96-well microplate reader (BioTek) [22]. Unstained cells were used as blanks. Fluorescence signal was corrected for variation in cell density by the ratio of fluorescence intensity to cell density OD_600_ [12,22]. The relative lipids intensity ratio after normalization was to reflect the difference among various cultures.

### 2.3. Lipid Droplet Observation

Yeast cells were first stained with Nile red as described above. Intracellular particles were then visualized by a fluorescence microscopy (Olympus BX51, Tokyo, Japan) [16]. Images were captured using cellSens Standard software.

### 2.4. Preparing Cells for Metabolome and Transcriptome

Yeast cells were cultured in 250-mL shake-flasks containing 50 mL of YPD liquid medium. After 12 h of incubation at 28 °C, the cultures were supplemented with 100 ng/mL rapamycin, and were further incubated for 12 h at 28 °C. The yeast cells were harvested by centrifugation at 9000× *g* for 5 min for metabolomics and transcriptomics analysis.

### 2.5. Intracellular Metabolites Extraction

The intracellular metabolites were extracted according to previous methods [23] with appropriate modifications. In brief, the obtained yeast cells (60 mg, wet mass) were transferred to a 2-mL centrifuge tube containing 500 μL methanol and 500 μL ddH_2_O. After vortex for 30 s, 100 mg glass beads were added into the tube. The tube was placed in liquid nitrogen for 5 min, thawed at room temperature, and ground for 2 min at 60 Hz in a high flux organization grinding apparatus. The cells debris were removed by centrifugation at 13,000× *g* for 10 min. The resulting supernatant was concentrated and left to dry in vacuum. The sample was dissolved in 300 μL 2-chlorobenzalanine (4 ppm) methanol aqueous solution (1:1) at 4 °C. After filtration with a 0.22 µm filter, the obtained samples were further investigated by LC-MS/MS.

### 2.6. LC-MS/MS Analysis

A Thermo Ultimate 3000 system equipped with an ACQUITY UPLC^®^ HSS T3 (150 × 2.1 mm, 1.8 µm, Waters, Milford, DE, USA) column was employed to separate the metabolites. The temperatures of autosampler and column were maintained at 8 °C and 40 °C, respectively. The sample volume was 2 µL. Mobile phase in negative-ion mode was solvent A (5 mM ammonium formate in water) and solvent B (acetonitrile). Mobile phase in positive-ion mode was solvent C (0.1% formic acid in water) and solvent D (0.1% formic acid in acetonitrile). The elution procedure was as follows: 2% B/D (0~1 min), 2%~50% B/D (1~9 min), 50%~98% B/D (9~12 min), 98% B/D (12~13.5 min), 98%~2% B/D (13.5~14 min), 2% D in positive model (14~20 min) or 2% B in negative model (14~17 min). The flow rate was set as 0.25 mL/min. The separated metabolites were then analyzed on a Thermo Q Exactive mass spectrometer with electrospray ionization (ESI) system [24]. The spray voltages applied were 3.8 kV in positive mode and 2.5 kV in negative mode. Sheath gas and auxiliary gas were set as 30 and 10 arbitrary units, respectively. The capillary temperature was maintained at 325 °C. The *m*/*z* scan range was 81 to 1000 for full scan at a mass resolution of 70,000 [24]. Data-dependent acquisition (DDA) and dynamic exclusion were performed according to the procedures as described [24,25].

### 2.7. RNA-Seq Analysis

Total RNA of the yeast strain 2E00681 was extracted using the Trizol Reagent (Invitrogen Life Technologies, Carlsbad, CA, USA). RNA quality was determined using a NanoDrop spectrophotometer (Thermo Scientific, Waltham, MA, USA). The mRNA was purified by poly-T oligo-attached magnetic beads followed by random fragmentation. The cDNA was subsequently synthesized according to the protocols as described [26]. After adenylation and ligation with Illumina PE adapter oligonucleotides, the cDNA fragments (400–500 bp) were purified using an AMPure XP system (Beckman Coulter, Beverly, MA, USA). The obtained cDNA fragments were then enriched by PCR using Illumina PCR Primer Cocktail. The resulting products were purified by AMPure XP system followed by quantitation using an Agilent high-sensitivity DNA assay on a Bioanalyzer 2100 system (Agilent, Santa Clara, CA, USA). The sequencing library was sequenced on a NovaSeq 6000 platform (Illumina, San Diego, CA, USA). Three replicates for each treatment were investigated in RNA-Seq experiments.

### 2.8. Bioinformatics Analysis

For MS/MS dataset, the obtained data were translated into mzXML format by Proteowizard (v3.0.8789) [27]. Peaks identification, peaks filtration and peaks alignment were performed using R XCMS package [24,28]. Quality control was performed as described [29]. All analytes were analyzed using hierarchical cluster [30]. Multivariate statistical analysis was performed using R language ropls package [31]. The differentially expressed metabolites were further investigated based on metabolomics pathway analysis (MetPA) (www.metaboanalyst.ca, accessed on 31 March 2020).

For the RNA-Seq dataset, after discarding adapter and low-quality reads, the filtered reads were mapped against the database of *Y. lipolytica* using TopHat [32]. HTSeq was employed to calculate the read count value of each gene [33]. Gene expression level was normalized using the reads per kilobase per million mapped reads (RPKM) [34]. DEGseq, an R package, was employed to calculate the differentially expressed genes [35]. GO (Gene Ontology, http://geneontology.org, accessed on 8 January 2020) and KEGG (Kyoto Encyclopedia of Genes and Genomes, http://www.kegg.jp, accessed on 8 January 2020) were used to annotate all the differentially expressed genes, respectively.

### 2.9. Fluorescent Real-time qPCR

Total RNA in yeast cells (about 100 mg, wet mass) was isolated using Trizol Reagent (Invitrogen Life Technologies) followed by synthesis cDNA using All-In-One RT MasterMix (abm) according to the manufacturer’s protocols. Fluorescent real-time qPCR was performed using SYBR Premix Ex Taq (Takara, Shiga, Japan). The primers for fluorescent real-time qPCR were shown in Appendix A. The 2^–ΔΔCt^ method was used to analyze relative mRNA expression levels [36].

## 3. Results

### 3.1. Effects of Rapamycin on Growth and Lipids Accumulation in Y. lipolytica under Nitrogen-Rich Conditions

YPD medium contains a large amount of yeast extract and peptone, which is rich in nitrogen sources with a total nitrogen concentration of 0.415%. The effects of rapamycin on growth and oils biosynthesis in the oleaginous yeast *Y. lipolytica* grown on YPD plate were investigated. As shown in Figure 1a, addition of rapamycin to YPD agar reduced the colony size and thickness of *Y. lipolytica*. We further counted the number of cells in the colonies and found that rapamycin did indeed reduce the oleaginous yeast cell growth (Figure 1b), which was consistent with the results in Figure 1a. However, the lipid fluorescence intensity in the cells treated by rapamycin was not significantly increased compared with that of untreated yeast cells in the oleaginous yeast *Y. lipolytica* (Figure 1c). Consistent with these results, the intracellular lipid droplets of the rapamycin-treated cells were similar to those of the nontreated cells (Figure 1d).

### 3.2. Overview of the Metabolome Profiling in Y. lipolytica Response to Rapamycin

To investigate how rapamycin affects intracellular metabolism in *Y. lipolytica*, we performed a metabolome analysis. After treating the yeast cells with or without rapamycin at 100 ng/mL for 12 h, the intracellular metabolites were extracted and analyzed by LC-MS/MS analysis. To monitor deviations of the data, principal component analysis (PCA) and relative standard deviation (RSD) of the quality control (QC) samples were investigated. As shown in Appendix A, the score plots of PCA in QC samples (red-marked) demonstrates good repeatability, and the proportion of characteristic peaks with RSD (<30%) exceed 70%, revealing that MS data meet the following experimental analysis requirements [37]. Finally, 176 differentially accumulated metabolites from the different groups were identified by mass spectrometry, of which 99 (56.2%) metabolites were up-regulated; 77 (43.8%) metabolites were down-regulated (Appendix A).

We further classified the obtained metabolites and found that the most dominant groups in the up-regulated metabolites were amino acids, peptides, and analogues (28%), followed by the unclassified component, null (26%) (Figure 2a). Among the down-regulated metabolites, the majorities were the unclassified components null (30%), and amino acids, peptides, and analogues (19%) (Figure 2b). Since various metabolites were involved in amino acid metabolism (Figure 2a,b), we further analyzed the levels of 20 amino acids that constitute proteins. After drug treatment, nine amino acids, tryptophan, isoleucine, proline, serine, glutamine, histidine, lysine, arginine and glutamic acid were up-regulated, while only three amino acids, namely threonine, tyrosine and aspartic acid were down-regulated (Figure 2c and Appendix A). Fatty acids were the major components of the oil particles in oleaginous yeast [19]. We found a 11.67-fold reduction in stearic acid (C_18:0_) and a 2.24-fold increase in stearidonic acid (C_18:4_) after the rapamycin treatment (Figure 2c and Appendix A). These results indicated that rapamycin altered the metabolic flows of amino acids and fatty acids in the oleaginous yeast *Y. lipolytica*.

### 3.3. Metabolic Pathway Analysis of the Obtained Metabolites

MetPA, which based on the KEGG metabolic pathway [38], was used to identify the relevant metabolic pathway in which the metabolites involved. Finally, a total of 48 metabolic pathways were obtained by MetPA (Appendix A). The top-10 important pathways impacted by the metabolites from MetPA analysis were shown in Figure 3a. As shown in Figure 3a, multiple metabolic pathways directly associated with amino acid metabolism. The most important metabolic pathway was alanine, aspartate and glutamate metabolism (Figure 3a), in which eight compounds were found to be significantly changed after treatment with rapamycin in *Y. lipolytica* (Figure 3b). Further analysis showed that three compounds, C03406 (argininosuccinic acid), C00064 (glutamine) and C00025 (glutamic acid) were up-regulated and five compounds, C00049 (aspartic acid), C03794 (adenylsuccinic acid), C00438 (carbamoylaspartate), C00334 (gamma-aminobutyric acid) and C00042 (succinic acid) were down-regulated. The main components in the lipids synthesized by *Y. lipolytica* are TAGs and SEs. However, compared with amino acid metabolism, the impacts of rapamycin on lipids metabolism were weaker according to MetPA (Appendix A). These results indicated that rapamycin affected multiple metabolic pathways of *Y. lipolytica*, especially amino acid metabolism.

### 3.4. RNA-Seq Data Analysis

To detect the differentially expressed genes in response to rapamycin by *Y. lipolytica*, an RNA-Seq analysis was performed. After DESeq analysis, we obtained 2224 differentially expressed genes (*p* < 0.05), in which 1142 genes were up-regulated (Appendix A) and 1082 genes were down-regulated (Appendix A). A volcano map of the differentially expressed genes was shown in Figure 4a. To verify the obtained genes in RNA-Seq, we used fluorescent real-time qPCR to detect the transcription level of 11 genes and found that the expression patterns of real-time qPCR and RNA-Seq were similar (Figure 4b), indicating that the RNA-Seq data had high reliability.

GO functional enrichment of all the differentially expressed genes was performed. Finally, 338 subgroups (*p* < 0.05) were obtained and many differentially expressed genes were associated with ribosome, catalytic activity, transferase activity, transferring one-carbon groups, and cofactor binding was highly enriched (Appendix A). Consistent with these findings, the genes related to metabolism and genetic information processing were significantly enriched base on KEGG pathway enrichment analysis (Appendix A). The top-20 enriched metabolic pathways were shown in Figure 4c. Among the top-20 pathways, seven pathways were related to amino acid metabolism, namely valine, leucine and isoleucine degradation; phenylalanine metabolism; alanine, aspartate and glutamate metabolism; tyrosine metabolism; valine, leucine and isoleucine biosynthesis; tryptophan metabolism; and arginine biosynthesis (Figure 4c). Two pathways were involved in lipids metabolism, namely fatty acid degradation and alpha-linolenic acid metabolism (Figure 4c). These observations further confirmed the ability of rapamycin to alter multiple metabolic processes at the transcriptome level in *Y. lipolytica*.

### 3.5. Correlation of Transcriptomics and Carbon Fluxes Involved in Lipids and Amino Acids

Although transcriptome data suggest a global transcriptional response to rapamycin by the oleaginous yeast *Y. lipolytica*, changes in transcriptional levels may not necessarily translate into changes in protein levels. Correspondingly, if these proteins are enzymes, then the level of downstream compounds catalyzed by the enzymes do not necessarily change. Therefore, the correlation of transcriptomic and metabolic fluxes should be further investigated. Based on metabolomic profile (Appendix A) and RNA-Seq analysis (Appendix A), many compounds and differentially expressed genes were related to amino acids and lipids. To uncover the relationship between transcriptome and metabolome, we reconstructed the biosynthetic pathways of intracellular lipids and amino acids (Figure 5). As shown in Figure 5, the transcriptions of *ACL1*, *ACL2*, *ACC*, *FAS1*, *FAS2* and *FABF*, which were involved in fatty acid biosynthesis, were down-regulated after rapamycin treatment in *Y. lipolytica*. However, many genes involved in oxidative degradation of fatty acids, such as *ACOX1*, *ACOX2*, *ACOX3* and *ACOX4*, were up-regulated (Figure 5). Another component of microbial oil droplets was steroids. As shown in Figure 5, when treated with rapamycin, the key genes *CYP51*, *ERG24*, and *ERG25* related to steroids synthesis were down-regulated. These observations suggested that rapamycin could inhibit the metabolic flow of lipids biosynthesis under nitrogen-rich conditions. In contrast, many genes involved in amino acid synthesis, such as *ILVE*, *ALT*, *GLNA1*, *GLNA2*, *GDHA*, *GUDB*, *ARG1*, *PROB*, *PCDH* and *PRODH*, were up-regulated after rapamycin treatment (Figure 5). Accordingly, multiple amino acids, including isoleucine, glutamine, arginine, glutamic acid and proline, were enhanced (Figure 5), indicating that the metabolic flows of several amino acids in the oleaginous yeast *Y. lipolytica* were changed when TOR was inhibited by rapamycin.

## 4. Discussion

Lipids accumulation in oleaginous yeasts requires nitrogen-starvation conditions [7]. Nitrogen deficiency inhibits the activity of TOR [11], indicating that the inhibition of TOR in oleaginous yeasts promotes lipids synthesis. TOR can also be passivated under rich nutrient conditions by rapamycin. A high level of intracellular lipids was observed when *Y. lipolytica* was cultured under a nutrient-deficiency condition (double limitation of nitrogen and magnesium) [39]. We found that rapamycin treatment of the oleaginous yeast *Y. lipolytica* did not significantly promote intracellular lipids biosynthesis under rich nutrient conditions. Lipids biosynthesis is also inhibited by rapamycin in mammals and mammalian TOR (mTOR) must be active to allow growth factors to induce lipogenesis synthesis [40]. Unlike the oleaginous yeast *Y. lipolytica*, TOR inhibition by rapamycin resulted in a fast lipids biosynthesis under nutrient-rich conditions in *S. cerevisiae* [12]. Two TOR complexes, TORC1 and TORC2, have been identified in *S. cerevisiae* and only TORC1 is rapamycin-sensitive [41], while only a single TOR kinase has been found in *Y. lipolytica* [17]. However, the effects of the number of TOR complexes on lipids biosynthesis in yeasts is still completely unknown.

CA is the initial compound that initiates lipids biosynthesis in oleaginous microorganisms, which was converted by ACL to AcCoA, resulting in excessive production of AcCoA, which is the major precursor of fatty acid biosynthesis [7]. As shown in the paper, the production of CA decreased by 19.59-fold after rapamycin treatment, indicating that the oleaginous yeast cells lacked sufficient substrate for oils biosynthesis. The oleaginous yeast *Y. lipolytica* could convert acetate into CA through the glyoxylate shunt pathway [42]. However, rapamycin did not change the levels of glyoxylate in this research. Heterologous expression of ACL from *Mus musculus* in *Y. lipolytica* could enhance lipids accumulation [43]. The oleaginous yeast *Y. lipolytica* owns two different subunits of ACL, Acl1p and Acl2p [15]. The genes *ACL1* and *ACL2* have been proved to play important roles in lipids synthesis [39,44]. These two core lipogenesis genes were also down-regulated by rapamycin. Fatty acid biosynthesis involves multiple reactions that convert AcCoA into long-chain fatty acids. Firstly, ACC catalyzes the carboxylation of AcCoA to malonyl-CoA (MalCoA). Then, AcCoA and MalCoA are condensed to acylCoA by FAS1 and FAS2 [8]. The genes *ACC*, *FAS1* and *FAS2* were down-regulated after rapamycin treatment in *Y. lipolytica*. These findings confirmed that rapamycin can reduce the carbon metabolic flux of lipids biosynthesis by inhibiting TOR activity in *Y. lipolytica*.

Inhibition of TOR by rapamycin altered amino acid metabolism of *S. cerevisiae*, especially arginine metabolism and glutamine metabolism [45]. Consistent with these observations in *S. cerevisiae* [45], rapamycin altered the metabolism of several amino acids, including arginine, proline, glutamine and glutamic acid in *Y. lipolytica*. In *S. cerevisiae*, a significantly low level of alpha-ketoglutaric acid (αKG) was coupled with high levels of alanine, arginine, asparagine, glutamic acid and glutamine [45]. In this study, rapamycin did not change the concentration of αKG and asparagine in *Y. lipolytica*, but increased the levels of arginine, glutamate and glutamine. These observations provide a possible industrial application for the amino acid production by *Y. lipolytica*. The biosynthesis of leucine has been demonstrated to be involved in lipids biosynthesis in *Y. lipolytica* [46], and leucine can act as a trigger that stimulates oils biosynthesis in *Y. lipolytica* [44]. We found that TOR inhibition increased the content of isoleucine but not leucine. These findings reinforce the notion that rapamycin can alter multiple metabolisms in the oleaginous yeast *Y. lipolytica*, including amino acid and oils metabolisms.

Nitrogen is one of the major factors influencing gene expression in *Y. lipolytica* [47]. GATA zinc-finger proteins were found to be involved in the regulation of nitrogen metabolism in *Y. lipolytica* [47,48]. The transcriptional level of gene *GLN3* (YALI0_D20482g), encoding a GATA transcription factor orthologue of Gln3 in *S. cerevisiae* [48], was increased by 32.7-fold. Protein function can be changed by phosphorylation [49]. It has been well documented that TOR restricts the entry of transcription factors, Gln3, Gat1, Msn2 and Msn4 into the nucleus by increasing the phosphorylation levels of these proteins under high nutrient conditions [11]. Both nutrient deficiency and addition of rapamycin inhibit TOR activity, leading to dephosphorylation of Gln3, Gat1, Msn2, and Msn4, which subsequently enter the nucleus [11]. However, there were no significant changes in the transcriptional level of genes *GAT1*, *MSN2*, and *MSN4* when TOR activity was inhibited by rapamycin in *Y. lipolytica*. After TOR inactivation, *GLN3* was significantly up-regulated, suggesting that Gln3 may be directly regulated by TOR. However, the role of Gln3 in amino acid and lipids metabolisms has to be investigated in the oleaginous yeast *Y. lipolytica*.

## 5. Conclusions

In conclusion, metabolomics and transcriptomics methods were used to identify the differentially accumulated metabolites and the associated key genes after rapamycin treatment in *Y. lipolytica* grown in nitrogen-rich medium. A total of 176 intracellular metabolites were obtained, including 12 amino acids (tryptophan, isoleucine, proline, serine, histidine, lysine, arginine, glutamine, glutamate, threonine, tyrosine and aspartate). Significant changes were also found in the contents of two fatty acids, stearic acid and stearidonic acid. Transcriptomic analysis showed that rapamycin down-regulated 11 genes related to lipids biosynthesis, and up-regulated 14 genes involved in fatty acid oxidation and amino acid biosynthesis. The results of this study are helpful to further study the physiological process of lipids accumulation in the oleaginous yeast *Y. lipolytica* grown in a nitrogen-rich environment.

## Figures and Tables

**Figure 1 jof-08-00939-f001:**
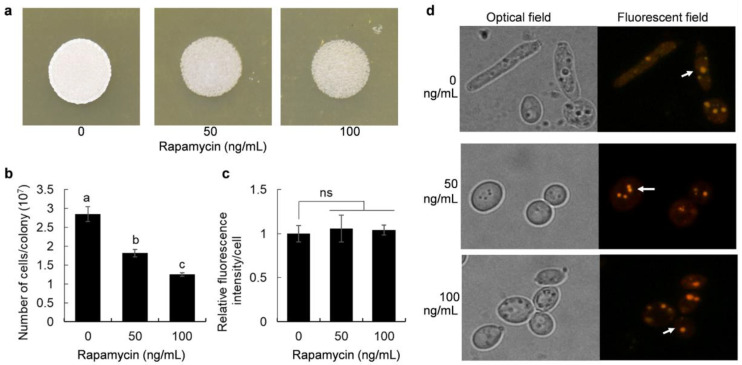
Impacts of rapamycin on cells growth and oils biosynthesis; (**a**) Colonies of the oleaginous yeast strain 2E00681 grown on YPD plates supplemented with different concentrations of rapamycin for 24 h; (**b**) Number of cells in the colonies; (**c**) Relative fluorescence intensity in cells; (**d**) Lipid droplets indicated by white arrows in the yeast strain 2E00681. Data are given as the mean ± SD, *n* = 3. One-way ANOVA was used to calculate the significant difference. Different letters (a, b, and c) marked on the charts are significant difference at *p* < 0.05. No significance was marked as ns.

**Figure 2 jof-08-00939-f002:**
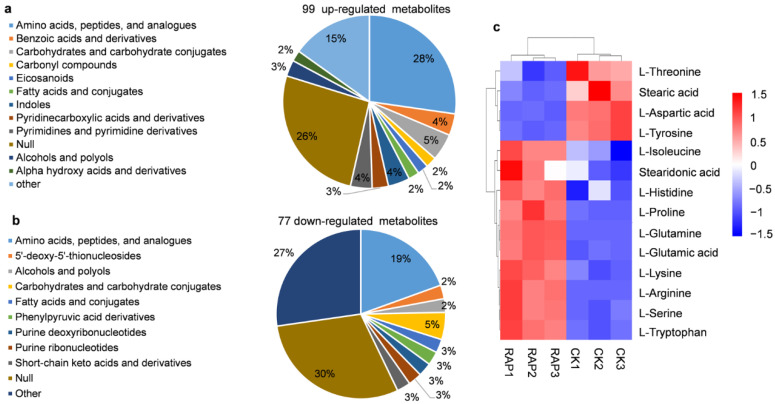
The metabolites identified in metabolomic analysis; (**a**) Classification of up-regulated metabolites; (**b**) Classification of down-regulated metabolites; (**c**) Heat map of amino acids and fatty acids. RAP means rapamycin treatment.

**Figure 3 jof-08-00939-f003:**
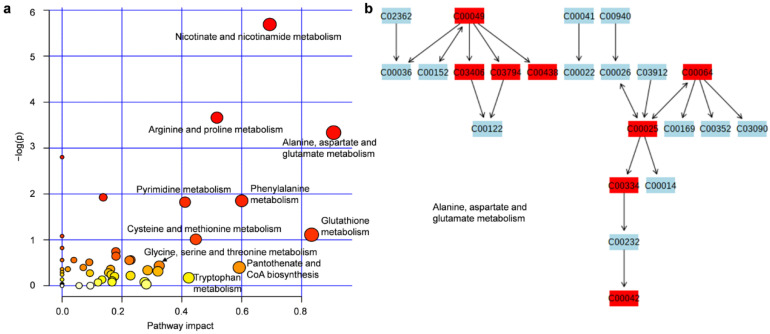
MetPA visualization diagram; (**a**) The metabolic pathways involved in the identified metabolites. Each dot means a metabolic pathway; (**b**) Alanine, aspartate and glutamate metabolism. Compounds ID were indicated in KEGG. The identified compounds were highlighted in red.

**Figure 4 jof-08-00939-f004:**
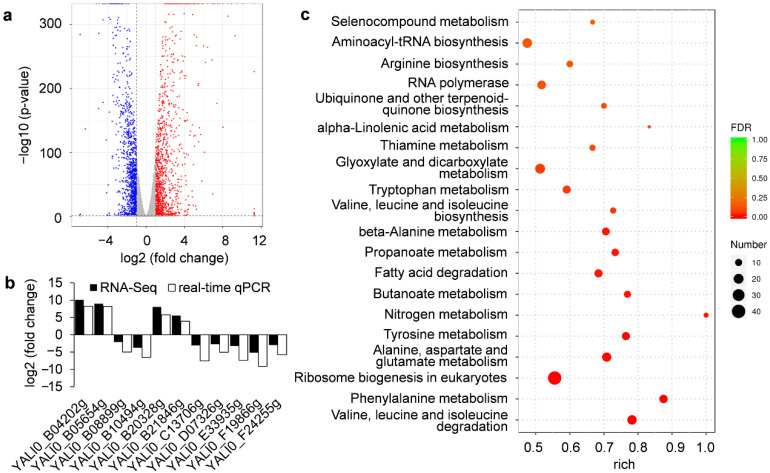
RNA profiling of *Y. lipolytica* in response to rapamycin; (**a**) A volcano map of the obtained genes; (**b**) Validation of RNA-Seq data using fluorescent real-time qPCR. The reference gene for normalization used in this study was 26S rRNA; (**c**) The top-20 enriched KEGG pathways in which the differentially expressed genes were involved after rapamycin treatment.

**Figure 5 jof-08-00939-f005:**
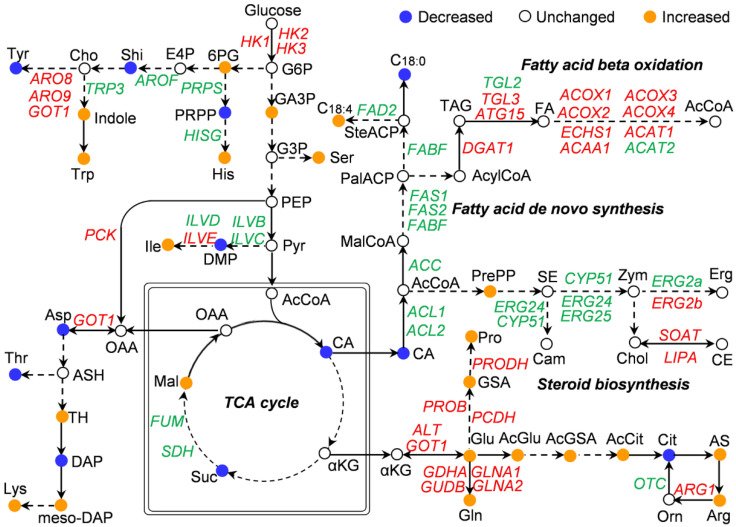
Schematic overview of lipids and amino acid changes in metabolic flux and transcript level. The up-regulated genes and the down-regulated genes were lighted in red and green, respectively. The colored balls (blue, brown, and white) represent different compounds. Abbreviations for enzyme and annotations are shown in Appendix A. Abbreviations for metabolites are listed in Appendix A.

## Data Availability

All data presented in the study are included within the article and its Appendix A or have been deposited in NCBI with accession PRJNA870251.

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
