# Peer review of "Metabolome and Transcriptome Profiling Reveal Carbon Metabolic Flux Changes in Yarrowia lipolytica Cells to Rapamycin"

_jof, 2022, doi:10.3390/jof8090939_

Round 1

Reviewer 1 Report

2.1. The genotypes of strains need to be indicated.

3.1. The method of counting the number of cells in the colonies needs to be described in Method section

In Discussion section  more detailed comparison with S. cerevisiae  would be helpful.

In Conclusion section  the sentence needs to be clarified: A total of 176 metabolites were 344 obtained, in which amino acids, peptides, and analogues were higher than other metabolites.

Reviewer 3 Report

In the paper entitled: “Comprehensive Metabolome and Transcriptome Profiling Reveal Carbon Metabolic Flux Changes of Yarrowia lipolytica to Rapamycin”, the authors entail to verify that inhibiting TOR activation benefits lipid production in oleaginous yeasts, herein, Y. lipolytica, the only documented species of the genus Yarrowia, which is ubiquitous in nature and has significant industrial value as well as importance in the food and medicinal industries. The results revealed that Rapamycin down-regulated several genes involved in lipid biosynthesis while up-regulating multiple genes involved in fatty acid oxidation and amino acid production, according to transcriptomic analysis (Rapamycin lowered carbon flux through lipid biosynthesis while increasing carbon flux through fatty acid breakdown and amino acid production (particularly glutamic acid, glutamine, proline, and arginine)). The results documented in this paper are interesting, however, some minor revisions are needed.

Strengths and Minor Corrections:

Strengths:

- The paper is well-written and easy to follow. The problem is well motivated, and the descriptions of the analytical experiments are very clear.

- The findings presented here are useful in studying the molecular mechanisms of amino acid and lipid metabolism in oleaginous yeast.

- Figure 5. Illustrates the work done by the authors perfectly, where the up-regulated and down-regulated genes involved in amino acid and lipids production.

Minor corrections:

- The paper skipped the possible industrial applications of Y. lipolytica treated with Rapamycin.

- There is no reference for Metabolites Extraction section.

- Some language deficiencies were detected, please rewrite the aim of the study (line 71-75), and the paragraph (Lines 315-327), to make them clearer.

Reviewer 4 Report

The proposed paper is well written and well structured, there are only a few things to correct.

The introduction could be expanded and more recent papers should be added in bibliography, some citations are a bit dated.

I would also suggest that the novelty of the proposed work be brought out more clearly.

Fig.4b is unclear

Very well the part of the discussions, little that of the conclusions that should be deepened
